# Interactive Effect of Biochar and Bio-Compost on Starting Growth and Physiologic Parameters of Argan

**Hassan El Moussaoui** [1,*] and **Laila Bouqbis** [2]

1   Laboratory of Biotechnology, Materials and Environment, Faculty of Sciences, Ibn Zohr University, Agadir 80000, Morocco
2   Laboratory of Biotechnology, Materials and Environment, Faculty of Applied Sciences, Ibn Zohr University, Agadir 80000, Morocco; l.bouqbis@uiz.ac.ma
*   Correspondence: hassan.elmoussaoui@edu.uiz.ac.ma

**Abstract:** The argan tree, which is found in southern Morocco, is characterized by environmental, economic and nutritional benefits, but the growth of this tree is very slow. This makes it necessary to find methods to accelerate its growth. A pot experiment was conducted to evaluate the effects of biochar (BC) and bio-compost (CP) each applied at the rate of 0, 3 and 6% (M/M) on starting growth of argan in fine silty soil for sixteen months. Main stem length, diameter, durability ratio, total length of all stems and number of sprouted shoots were measured every two months with two photosynthetic measurements spaced five months apart for each argan seedling. Despite the strong signs of epigenetic sensitivity and genetic variability across the argan behavior of each treatment depending on the duration and environmental conditions of the crop and the large standard deviations marked in all the tests that were conducted on the argan, some treatments showed interesting results, even in terms of the interaction between climatic conditions, type of treatment and type of test. The argan plants which were grown in the substrate at 6% BC 3% CP showed significant results for all the growth parameters studied and throughout the test. This mixture marked an average water holding capacity (WHC) of around 0.66 g $H_2O$/g dw; the argan seedlings showed the best perimeter average, which exceeded 2.7 cm in the last measurement, with a ratio (height/diameter) strictly less than 7, which removes any possible problem of argan filiform. However, argan plants from all treatments were not stable in the growth characteristics studied; each treatment has advantages and disadvantages regarding argan. Transplantation and monitoring in the field of argan seedlings that have had interesting results are strongly recommended to see if the good starting growth influences their development in the field or if it is a temporary effect.

**Keywords:** argan; biochar; bio-compost; starting growth





## 1. Introduction

The argan tree "*Argania spinosa*" is generally a plant that exists exclusively in the Souss plain in Morocco. An oil of high nutritional value, "high level in unsaturated fatty acids", cosmetic, environmental and economic value, pastoral plant and a better quality wood are the main characteristics of this tree [1–3]. Although the argan tree has a long lifespan (several hundred years), its growth is very slow, and it may take decades to reach its maturity stage [4–7]. The type, nature and composition of the substrate directly influence forest productivity, especially the upper part of the substrate, which provides and maintains all the elements necessary for plants growth and productivity [8,9]. Plants can adjust their physiological and biochemical responses according to the composition of the base [10,11]. Soil amendment with organic fertilizers has significant economic and environmental benefits [12–14]. Biochar, the solid product of biomass pyrolysis, has been produced and utilized for several thousand years [15]. Biochar has been shown to have positive effects on effective sequestration of applied carbon and mitigating anthropogenic $CO_2$ emissions [16],

improves soil aggregate stability and plant available water [17], increases soil water holding capacity [18], increases adsorption efficiency of soil HMs [19], improves the soil physical environment [20] and enhances the productivity and soil enzymatic activities [21]. The type, dose and conditions of pyrolysis can cause negative effects on the soil and the plant. The authors of [20,21] showed that a high dose of biochar can increase soil alkalinity or lead to a high soil pH resulting in decreased nutrient availability and potentially Na toxicity, which causes reduced plant growth. Biochar volatile organic substances can also have deleterious effects on plant growth [22,23]. These negative effects can be overcome by mixing with compost, which increases soil fertility, uptake and plant productivity [24–26]. Bio-compost also has several beneficial effects on soil, productivity, quality and crop yield in several studies [27,28]. Pot studies have shown that the application of organic fertilizers can increase the quality of argan seedlings in the nursery, including the application of compost, which has shown interesting results in the quality and productivity of argan [29]. Careful selection of feedstock is crucial as the feedstock must be abundantly available, inexpensive, unavoidable and have a low embedded impact [26]. Biochar has been studied many times before, but not commonly on the argan tree. Studies on the growth of the argan tree are very limited, no study has been found that deals with the subject of starting growth of argan by applying the biochar either in pots or in the field, which makes our study a basis and an initiation of research in this direction. The objective of this study is to describe the physicochemical compositions of one type of bio-compost and a mixture of two biochars and their effects on soil CEC, cation-exchange capacity, and WHC at different concentrations and mixtures. It also aims at studying the behavior of argan plants grown in pots (to control the system given its sensitivity to changes in the culture medium) containing different doses of biochar and bio-compost. In addition, it aims to test the effect of the substrate on the starting growth of argan in pots before transplanting them to the field.

## 2. Materiel and Method

### 2.1. Soil, Biochar, Bio-Compost

Fine silty soil (33.77% sand, 51.88% silt and 14.35% clay) was recovered from an argan forest in southern Morocco. A mixture of two biochars "RAMAR" (75% biochar of municipal sewage "RAM" and 25% biochar of argan tree shell "AR") was prepared at a temperature around 450 °C Table 1. Regarding the bio-compost used in this study, it was purchased (5% humicogenic bacterial starter + 95% vegetable matter (grape marc, bagasses, pulps and fibers)).

**Table 1.** Conditions of pyrolysis of each biomass.

| Feedstock | Temperature | Rate | Residence Time | Yield |
|---|---|---|---|---|
| **RAM** (Biochar of municipal sewage) | 483 °C | 0.8 Kg | 55 min | 60% |
| **AR** (Biochar of argan tree shell) | 443 °C | 2 Kg | 30 min | 17% |

### 2.2. Water Holding Capacity

A pot with a volume of 22 cl, covered from below with a net and filter paper, was filled with two-thirds of the mixture to be tested and placed in a box filled with water and covered with aluminum foil for 24 h. After 24 h, the water was emptied out and the pots were left to drain for another 24 h; after draining, the WHC was calculated as g $H_2O$/g dw of each mixture to be tested.

### 2.3. Monitoring the Starting of Growth and Productivity of the Argan Tree

The pots test, at three repetitions for 16 months, from December 2019 to March 2021, in the open air without a greenhouse was conducted for 9 treatments with the soil as control. The treatments are composed of three concentrations, 0, 3 and 6%, in all possible combinations between biochar and bio-compost. This test was applied to 3-month-old

argan plants; growth and productivity were deduced from three parameters measured every two months: The first was the measurement of the length of the main stem in order to assess the effect of the type and composition of the substrate on the argan plant growth under different treatments. The productivity of the argan tree was evaluated through the sum of all the stems of each argan plant, and in order to assess the favorability of the growth of each treatment, we counted the number of buds generated every two months during the test period "ramification". The root collar diameter was also measured during the test to evaluate the vigor of argan under different treatments. The physiological parameters were evaluated through a IRGA photosynthesis test, which was carried out through ADC BioScientific LCi-SD System Serial No.33774. Three repetitions are made for each treatment; two IRGA measurements were taken during the test period spaced five months apart. The results of the photosynthetic rate, A ($\mu$mol $CO_2$ $m^{-2}$ $s^{-1}$), were measured. The temperature and humidity were recorded continuously every 10 min throughout the duration of the test via «UNI-T UT330B USB Testeur Humidity/thermometer Datalogger IP67» Figure 1.

### 2.4. BC, Bio-Compost and Soil Chemicals Analysis

The chemical contents of the soil, biochar and bio-compost were determined using a flame emission spectrophotometer for total Ca, Mg, K and Na. Fe, Mn, Zn and Cu content was taken using an atomic absorption spectrophotometer Thermo Scientific iCE 3000 series [30]. The $KH_2PO_4$ and $NaNO_3$ were measured using colorimetrical analyses [31]. The total soil organic carbon content was measured using the Walkley-Black chromic acid wet oxidation method [32], and the total nitrogen (TN) content was measured using the Kjeldahl method [33]. Total organic matter (TOM) and organic carbon (OC) were estimated from calcination, and phosphorus was determined via the OLSSEN protocol [34]. For CEC, different mixtures were prepared between soil, biochar and bio-compost with 70% WHC, and the mixtures were incubated in darkness for 4 weeks before the cation exchange capacity analysis. The CEC was determined by means of the ammonium acetate method (Metson method). The exchangeable bases were leached from the mixture with 1 M ammonium acetate at pH 7.0 and the excess removed by washing with alcohol, leaving the exchange sites saturated with ammonium ions. These were removed by leaching with M sodium chloride, and the CEC was determined by the amount of ammonium ions in the extract [35].

### 2.5. Data Analysis

For all tests of effects of different treatments, additions on all replicated measurements were tested via ANOVA. The significance of differences among treatment groups was determined with the Tukey test. A result was considered significant at $p < 0.05$. For the quantitative variables, we used the PCA method. All statistical tests were performed with SigmaPlot 4.5 (St. Louis, MO, USA; Systat Inc., 2020).

- Summary of Experiment Scheme

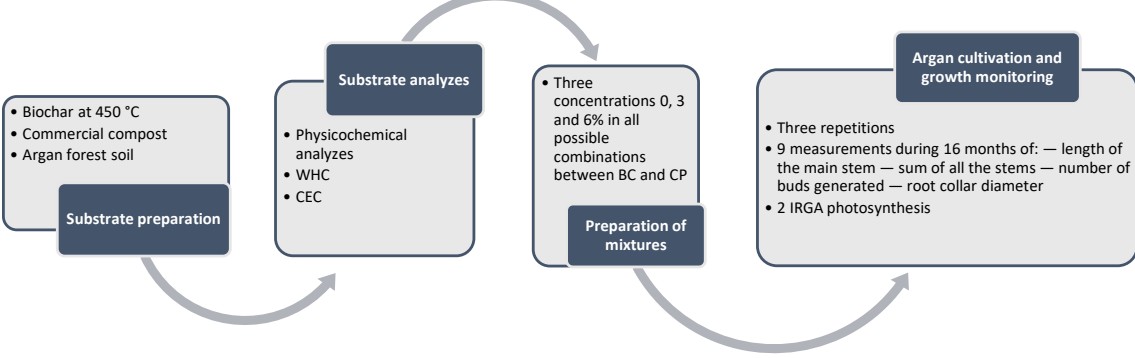

**Figure 1.** Summary of experiment scheme.

## 3. Results and Discussion

### 3.1. Correlations between the Different Parameters PCA

The PCA analysis (Figure 2) shows no correlation among the CEC and the average of the main stem perimeter and the number of buds. While there is a negative correlation between the CEC and the average total root length, main stem length and WHC, there is a positive correlation between the number of buds and the average perimeter of the main stem. The perimeter generally increases with decreasing WHC and has a partial correlation with the total length of argan.

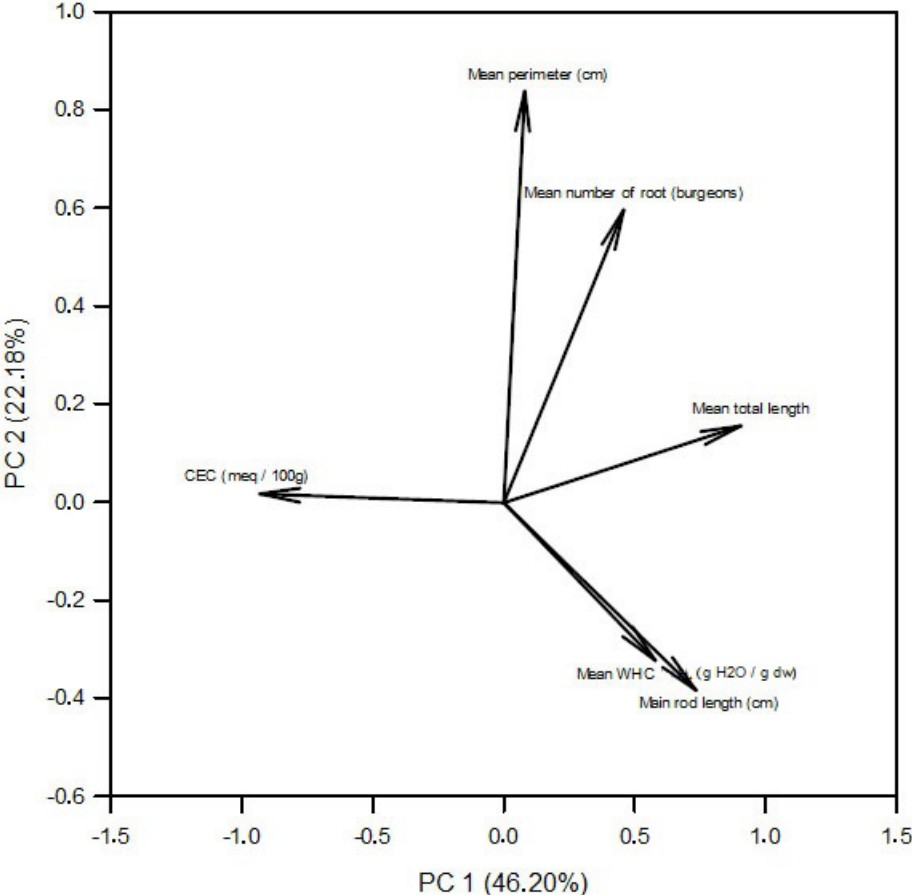

**Figure 2.** Biplot graph of the principal component analysis (PCA) for the variation of the following quantitative variables: WHC average (g $H_2O$/g dw); CEC (meq/100 g); perimeter average (cm); number of roots (burgeons); average main rod length (cm); total length. ▶ Variable.

Mohamed et al. (2016) [36] found that there was a strong positive correlation between CEC and WHC in all the mixtures tested. On the other hand, [36] (Cooper et al., 2020) did not find any correlation between CEC and WHC in all mixtures between biochar and compost at different doses [37]. This diversification of results between studies shows that the type and origin of biochar and/or compost directly determines WHC and CEC and the interaction between them. In addition, the type of soil with which they were mixed may also influence the CEC and WHC of the mixture and even the correlations between these two parameters. Our results show that the CEC increase has negative effects on some productivity parameters studied and has no relation to others. On the other hand, the ordinary effects of a high CEC increases soil fertility [38], which increases productivity on all levels [39–42]. A strong correlation was detected between the length and the diameter of argan in several ex situ tests [42]. These results show that the behavior of argan via the compositions and the physical chemical characteristics of soil is totally different from other plant species.

### 3.2. Water Holding Capacity

Figure 3 shows that the addition of biochar does not cause any significant difference between the different treatments although increasing the dose of biochar increases the mixing WHC. On the other hand, the 6% bio-compost significantly increases the WHC of soil compared to the other treatments. Generally, the WHC of these three treatments was around 0.7 g $H_2O$/g dw.

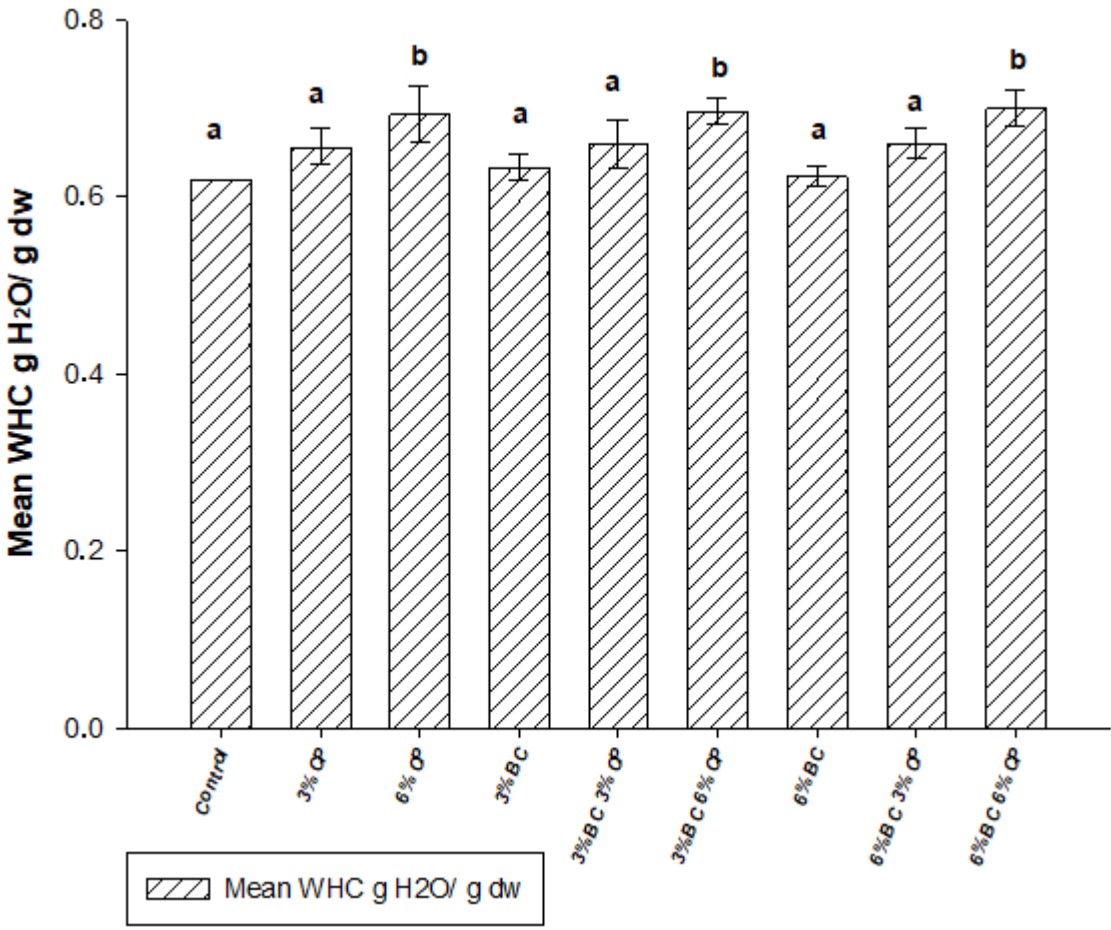

**Figure 3.** WHC average of different doses between biochar and bio-compost + standard deviation (*n* = 3); different superscript letters "a" and "b" represent significant differences between treatments at the *p* < 0.05 level; LSD value = 0.03. BC = biochar; CP = bio-compost.

The texture, structure and composition of soil directly influence the soil WHC, and the effect of biochar is clearly visible on sandy textured soils [43–45]. In addition, the effect of BC on WHC is clearly visible on sandy loam soil [45]. The biochar of different sources added to soil increases the water-holding capacity and might increase the water available for crop use [46,47]. The addition of compost increased the water holding capacity [48–50]. However, the synergy between compost and biochar increased the WHC more than their effect when separated [40,51].

### 3.3. BC, Bio-Compost and Soil Chemicals Analysis

The first table shows the effect of bio-compost and biochar in the attenuation of CEC. CEC generally decreases with the presence of biochar and/or bio-compost. Control at 0% BC 0% CP has the greatest CEC versus all other treatments at 9.99 Meq/100 g; the lowest CEC is detected in the mixture containing biochar "3% and/or 6%" and bio-compost at 6% at 5.3 Meq/100 g. No change in CEC value was seen between biochar at 3% bio-compost at

0% and biochar at 3% bio-compost at 3% and also between biochar at 6% bio-compost at 0% and biochar at 3% bio-compost at 3%.

The application of composted biochar made from seafood shell powder, peanut shell, commercial humate and inorganic nutrients on coastal soils increases CEC with increasing application rates [51]. Furthermore, the application of biochar based on waste willow wood (Salix spp.) or compost or both mixed on the Ferralsol increases the CEC [26,52,53]. The application of biochar to the sand had no significant effects on CEC. However, mixing between the compost and biochar significantly increased the CEC of sand [53].

Chemical soil analyses show the weakness of fine silty textured soils in terms of total organic matter, organic carbon and total nitrogen, 2.84, 1.65 and 0.059%, respectively. Minerals and trace elements differ from element to element. The compositions of biochar are widely different from bio-compost; this is mainly due to the initial biomass and the initial conditions of production. From a general perspective, bio-compost is characterized by a high rate of total organic matter, total organic carbon and total nitrogen compared to biochar. Furthermore, the rate of microelements is higher in bio-compost compared to biochar except iron. On the other hand, the rates of macro- and microelements are higher in biochar compared to bio-compost, except sodium, which has a slightly higher rate in bio-compost. Biochar RAMAR is characterized by a high level of magnesium and total phosphorus, 1.69 and 0.95%, respectively. In addition, high levels of copper and zinc are detected in bio-compost, 993.8 and 241.4, respectively.

The properties of biochar depend greatly on the nature of the biomass, the chemical composition and the production conditions used to produce biochar [54]. Conocarpus wastes that were pyrolyzed at a high temperature increased the total content of N, C, P, K, Ca and Mg [55]. Feedstock and temperature are critical parameters in determining the chemical composition of biochar. The carbon content was 392, 804 and 817 g kg$^{-1}$ at 500 °C for biochar made from poultry, litter peanut hulls and pine chips, respectively. The nitrogen content was 34.7, 24.3 and 2.55 g kg$^{-1}$ at 400 °C for biochar made from poultry, litter peanut hulls and pine chips, respectively. The total micronutrients were even influenced by the type of feedstock and temperature; copper was 1034, 19 and 9 mg kg$^{-1}$ at 500 °C for biochar made from poultry, litter peanut hulls and pine chips, respectively [56]. Furthermore, the compost compositions vary depending on the biomass and the processes used to make compost [57]. The maturity of compost was partly influenced by the C:N ratio content and method used to prepare the compost. Compost made from soybean residue and leaf litter was richer in nutrients than corn residue, weed biomass and urban waste [58] (Tables 2 and 3).

**Table 2.** Cation exchange capacity for all treatments.

| Treatement | Control 0% BC 0% CP | 0% BC 3% CP | 0% BC 6% CP | 3% BC 0% CP | 3% BC 3% CP | 3% BC 6% CP | 6% BC 0% CP | 6% BC 3% CP | 6% BC 6% CP |
|---|---|---|---|---|---|---|---|---|---|
| CEC (meq/100 g) | 9.99 | 9.16 | 8.49 | 6.66 | 6.66 | 5.83 | 6.83 | 6.827 | 5.828 |

**Table 3.** Biochar, soil and bio-compost chemicals analysis.

| | PH | CE µS/Cm | WHC g H$_2$O/g dw | % TOM (Organic Matter Content) | % OC (Organic Carbon) | % Nt (Total Nitrogen) | C/N | % Pt (Total Phosphorus) | % K | % Na | % Ca | % Mg | Fe mg/kg | Mn mg/kg | Cu mg/kg | Zn mg/kg |
|---|---|---|---|---|---|---|---|---|---|---|---|---|---|---|---|---|
| bio-compost | 8.01 | 2990 | 0.73 | 41.47 | 24.05 | 2.84 | 28.17 | 0.20 | 0.16 | 0.36 | 6.01 | 0.42 | 6079 | 232.5 | 993.8 | 241.4 |
| biochar RA-MAR | 9.40 | 3050 | 1.41 | 28.23 | 16.38 | 0.85 | 19.27 | 0.95 | 0.26 | 0.27 | 9.14 | 1.69 | 7482.3 | 217.1 | 196.5 | 59.0 |
| | | | | | | | | P$_2$O$_5$ 0/00 | K$_2$O 0/00 | Na$_2$O 0/00 | CaO 0/00 | MgO 0/00 | | | | |
| Soil | 8.49 | 60 | 0.62 | 2.84 | 1.65 | 0.059 | 28.06 | 0.341 | 1.711 | 0.503 | 4.978 | 1.221 | 1.1 | 13.8 | 0.5 | 1.8 |

### 3.4. Monitoring the Starting of Growth and Productivity of the Argan Tree

From a general view of Figures 4–7, it can be seen that the growth and physiological parameters studied are not linked only to the substrate in which it is cultivated but also to the stage of growth and to environmental conditions, since we see in all the treatments that the growth and the productivity of argan are not stable over time and their behavior over time is not anticipated. In addition, the genetic variability is clearly visible on all scales despite the fact that all the argan plants have the same provenance due to the detection of large standard deviations in the same treatment. The combination of these first two results may clarify the strong possibility of argan's susceptibility to an epigenetic effect.

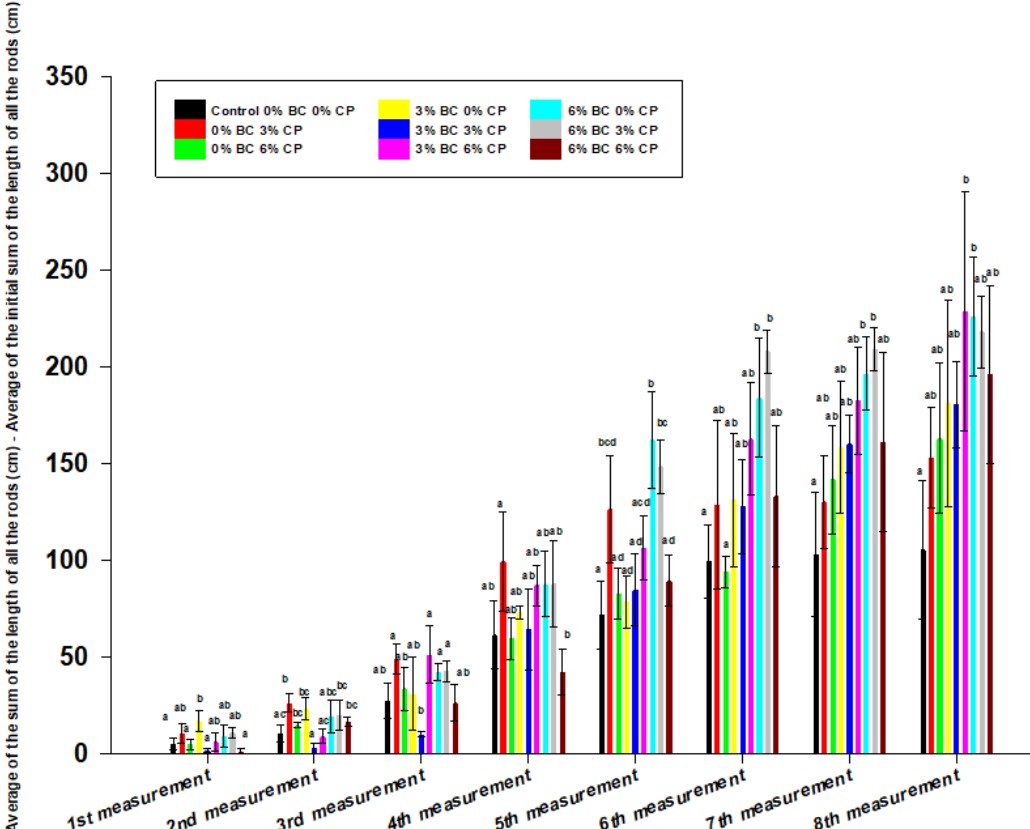

**Figure 4.** Average of the sum of the length of all the rods (cm)—average of the initial sum of the length of all the rods (cm) + standard deviation (n = 3). Different superscript letters "a", "b", "c" and "d" represent significant differences between treatments at the $p < 0.05$ level (a single common letter between two treatments means that there is no significant difference between these two treatments). ANOVA tests were made for each measurement separate from the others. BC = biochar, CP = bio-compost.

Figure 4 shows that in all the measurements the control remains the weakest compared to the other treatments at the level of the sum of the length of the totality of the stems of each argan plant. In the first 6 months, the treatment 0% BC 3% CP gives the best productivity. Then, this productivity decreases over time; the treatments 0% BC 6% CP, 3% BC 0% CP and 3% BC 3% CP bring stable productivity over time, from the 10th month onwards. Treatments 3% BC 6% CP, 6% BC 0% CP and 6% BC 3% CP remain the best treatments in terms of productivity. The 6% BC 6% CP treatment marks increasing productivity throughout the duration of the test. Although the perimeter of argan cultivated in the two treatments 0% BC 3% CP and 6% BC 3% CP is not statistically significant, it remains the best compared to all other treatments in all stages of growth and throughout the test duration (Figure 5).

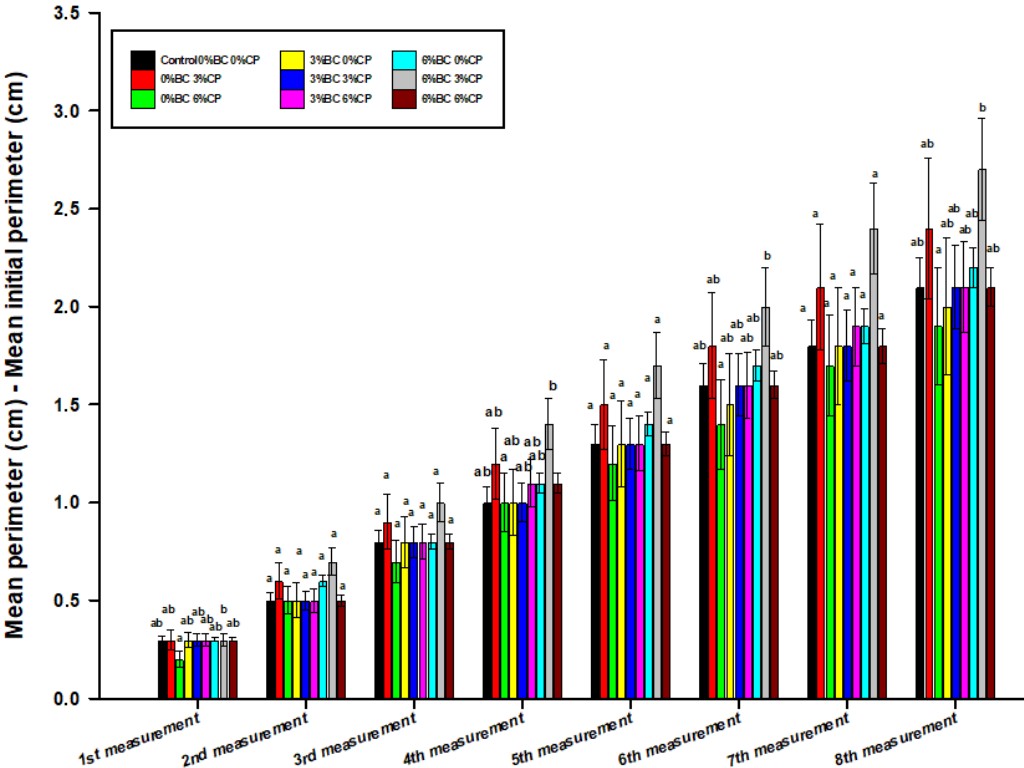

**Figure 5.** Mean perimeter (cm)—mean initial perimeter (cm) + standard deviation (n = 3). Different superscript letters "a" and "b" represent significant differences between treatments at the $p < 0.05$ level (a single common letter between two treatments means that there is no significant difference between these two treatments). ANOVA tests were made for each measurement separate from the others. BC = biochar, CP = bio-compost.

Figure 6 combines the study of two parameters at the same time: the length of the main stem and the number of stems. From November 2020, there was a sudden increase in the length of the main stem of two treatments, 3% BC 0% CP and 6% BC 6% CP, and the greatest lengths averaged 32.3 cm and 31 cm, respectively. On the other hand, the 6% BC 3% CP treatment maintains stable growth at the level of the main stem in each measurement. The 3% BC 3% CP and 0% BC 3% CP treatments recorded shorter main stem lengths than the control. Regarding the number of buds that emerged, stability was detected during all the measurements. Generally, the treatments 0% BC 3% CP, 3% BC 0% CP and 3% BC 6% CP recorded the best numbers of emerged buds; the number recorded in the last measurement was around 15 buds. Regarding the photosynthetic rate "A" of the first measurement, Table 4 shows that the change of substrate affects the photosynthetic rate of argan since it is observed that the largest A was marked in the control and 3% BC 3% CP, by an average of 13.33; 17.32 µmol $CO_2$ m$^{-2}$ s$^{-1}$, respectively. Due to the lack of time to adapt to the substrate, some other treatments scored very good "A" such as 3% BC 6% CP and 0% BC 6% CP, by 10.98 and 10.11 µmol $CO_2$ m$^{-2}$ s$^{-1}$, respectively. In the second measure, the treatment 3% BC 6% CP kept its high "A"; on the other hand, for the other treatments, the weakest in the two measures for A generally were marked for the treatment without BC and 6% BC 0% CP. No significant results were noted for the other treatments. The months which gave the best general productivity and which are the best and most suitable for transplanting argan to the ground differ from one treatment to another. One year is the best duration for argan plants for the control and the treatment 0% BC 3% CP in pots before transplanting them to the field. Concerning the treatment 6% BC 6% CP, this duration is not yet determined since the productivity increases with the increase of duration in pots. For the other treatments, the best duration in pots before transplanting them to the field is 14 months.

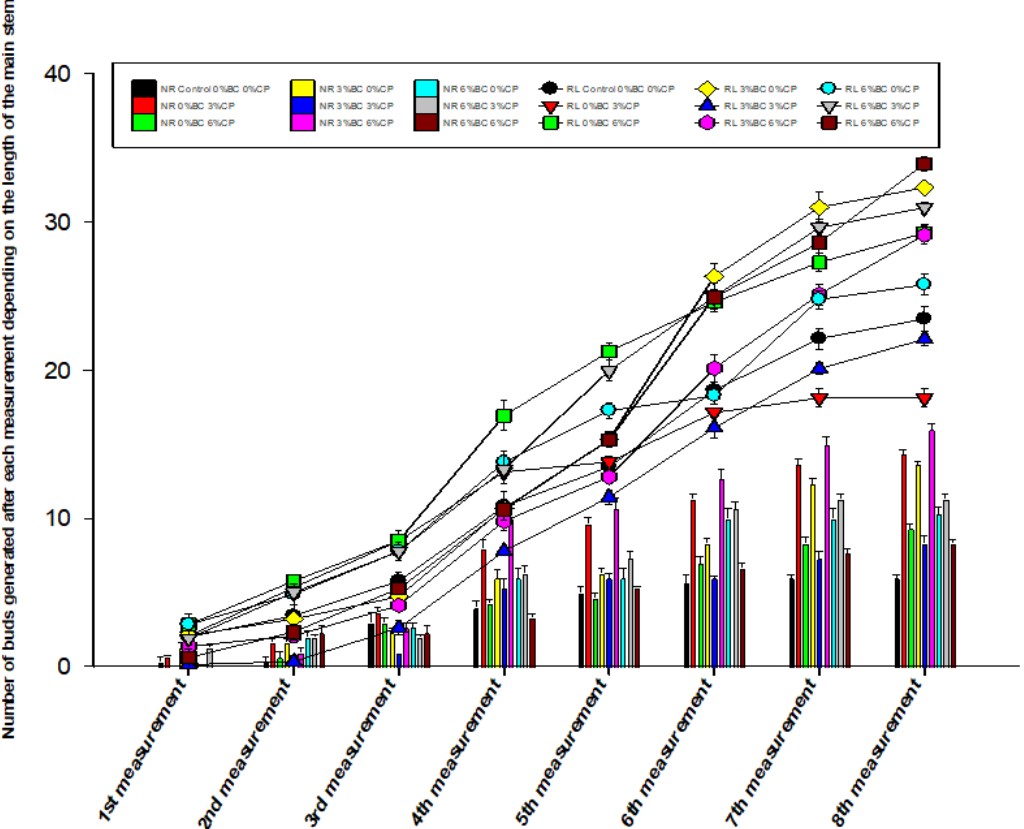

**Figure 6.** Number of buds generated after each measurement depending on the length of the main stem (cm), replication (n = 3). BC = biochar, CP = bio-compost, NR = number of buds, RL = length stem.

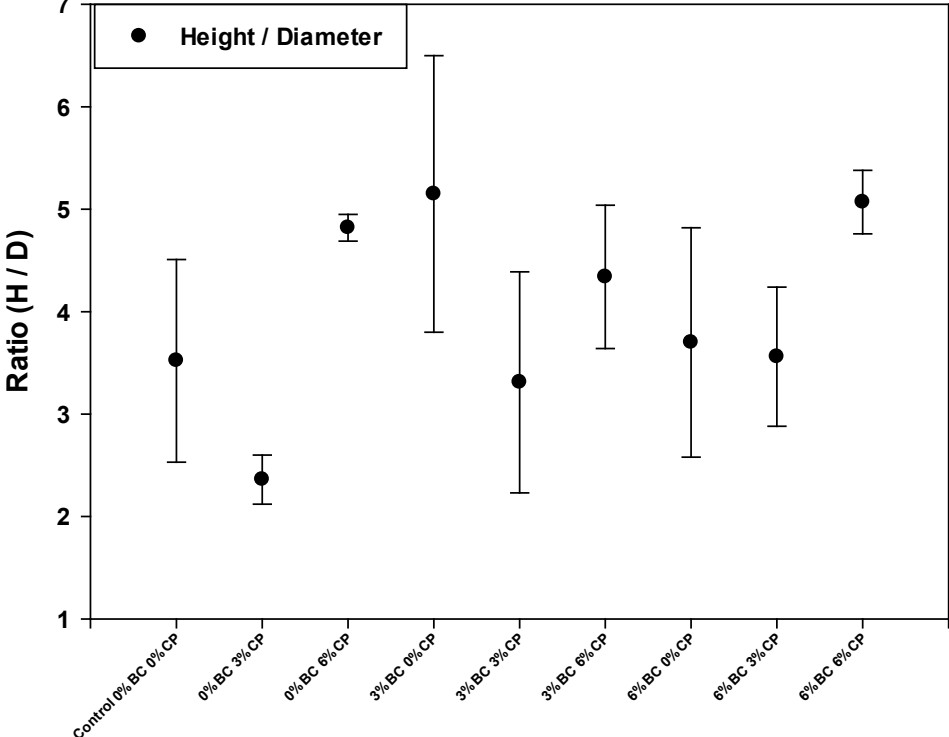

**Figure 7.** Argan height/diameter ratio (H/D).

**Table 4.** IRGA parameter; photosynthetic rate, A ($\mu$mol $CO_2$ m$^{-2}$ s$^{-1}$) for 2 measures of argan leaf + standard deviation (n = 3); different superscript letters represent significant differences between treatments at the $p < 0.05$ level. ANOVA tests were made for each measurement separate from the others. BC = biochar, CP = bio-compost.

| Treatement | Control 0% BC 0% CP | 0% BC 3% CP | 0% BC 6% CP | 3% BC 0% CP | 3% BC 3% CP | 3% BC 6% CP | 6% BC 0% CP | 6% BC 3% CP | 6% BC 6% CP |
|---|---|---|---|---|---|---|---|---|---|
| Measure Number 1st Measure Photosynthetic rate, A ($\mu$mol $CO_2$ m$^{-2}$ s$^{-1}$) | 13.33 ± 0.54 [c] | 6.31 ± 0.30 [ab] | 10.11 ± 0.54 [a] | 5.55 ± 2.8 [ab] | 17.32 ± 1.59 [cd] | 10.98 ± 3.32 [a] | 3.52 ± 2.38 [b] | 7.63 ± 2.54 [ab] | 9.05 ± 2.04 [ab] |
| 2nd Measure Photosynthetic rate, A ($\mu$mol $CO_2$ m$^{-2}$ s$^{-1}$) | 4.78 ± 1.7 [a] | 3.26 ± 1.31 [a] | 2.7 ± 1.14 [a] | 7.69 ± 2.73 [ab] | 7.31 ± 3.41 [ab] | 13.42 ± 3.42 [b] | 3.54 ± 0.66 [a] | 6.55 ± 2.9 [a] | 7.73 ± 0.94 [ab] |

The ratio height/diameter (Figure 7) shows that all the argan seedlings have good robustness. Despite the change in treatment, this ratio remains strictly less than 7 which removes any possible problem of argan filiform.

The argan tree [*Argania spinosa* (L.)] as a spontaneous and endemic xerophilic species, wild and forming well-developed forests in the Souss plain in southwestern Morocco [59], is characterized by high genetic variability [60]. Despite their resistance to different climatic conditions over time [61], according to our study, we see that argan behaves differently in terms of growth depending on the climate, the substrate and the parameter studied although the argan seedlings have only one origin. The various argan growth parameters are not stable during the culture period. The duration of argan tree breeding in the nursery, before its transplantation in the field, plays a very important role in the growth, in particular the height. Seventy-two months after transplanting, the argan seedlings which were transplanted after one year in breeding marked very low heights compared to those that were two years old [42]. Behavior and growth differ from region to region depending on the soil type and the climate [42]. Generally, height, diameter, leaf number and chlorophyll content of argan decrease with increasing NaCl dose, and the mortality rate increases [62]. A test of different substrates (peat, compost and potting soil) at different doses showed that the type, nature and composition of substrates play a key role in determining the length, diameter and dry weight of argan [29]. The substrate introduced a highly significant effect on the height and the diameter of the collar and the height/diameter ratio. A ratio greater than 8 means that the risk of the generation of an argan filiform is highly possible. It is also a bad indicator of the capacity of argan to overcome transplant shocks [29]. No significant effect of NaCl on argan was detected on the height/diameter ratio [62]. Regarding the photosynthetic activity of argan, it is generally not affected even under the conditions of water stress, which directly affects the photosynthetic activity of most of the plants [63]. The number of buds and plant branches is mainly related to environmental conditions including temperature and light [64–68].

The application of biochar on forest soil significantly increases the various parameters of soil fertility [67], which can increase the productivity of forests if it has been applied in a reasonable way; otherwise, the effect will be reversed [68]. The biochar positively influences the height, the diameter of the growth of the species and the final number of leaves, but this influence differs depending on the nature of the biochar, dose and also the species to which it has been applied [69]. Biochar or the combination of municipal green waste compost and biochar did not give any additional positive effect on the growth of Corymbia maculata compared to municipal green waste compost alone [70]. Without constrained conditions for plant growth (e.g., water stress, toxicity and nutrient deficiencies due to excessive leaching), the effect of biochar remains minimal to negligible [71]. Biochar and compost have largely positive effects on the growth of two tree species (*Samanea saman* and *Suregada multiflora*) [72]. Although the test was done in pots, the results were very unsteady

throughout the duration of the experiment. This is due to the high sensitivity of this species to changes in the culture medium and the high genetic variability. Refs. [61,73–76] have shown that argan is marked by a very large genetic variability even if the trees live in the same environment, which makes the behaviors and characteristics of this tree unstable or unexpected. The argan woodland ecosystem shows high climatic sensitivity [75]. This space consists of fragile argan ecosystems and accentuated ecological variation and is an open, complex and multifunctional system [76].

## 4. Conclusions and Perspectives

The nature, origin and conditions of production of biochar determine their compositions and their effects on the soil to which they are applied. This study showed that productivity, vigor and robustness are directly related to growth stage, genetic variability and environmental conditions. Despite that, the argan seedlings grown in soil amended with biochar and biocompost gave better results in most of the parameters studied compared to the control. The WHC of all 6% biocompost mixes was the highest compared to the other mixes. Chemical analyses of soil, biochar and biocompost have shown that the three have a basic PH more than 8 and that the microelements are very high in biochar and bio-compost. A prospective and additional study in the field is necessary to clarify the behavior of argan regarding the base and the substrate in which it is cultivated.

**Author Contributions:** Conceptualization, L.B.; Data curation, H.E.M.; Formal analysis, H.E.M.; Funding acquisition, L.B.; Investigation, H.E.M.; Methodology, L.B.; Project administration, L.B.; Resources, H.E.M.; Software, H.E.M.; Supervision, L.B.; Validation, L.B.; Visualization, H.E.M. and L.B.; Writing—original draft, H.E.M.; Writing—review & editing, L.B. All authors have read and agreed to the published version of the manuscript.

**Funding:** This research was funded by Laboratory of Biotechnology, Materials and Environment of the Faculty of Sciences of Agadir, the Polydisciplinary Faculty of Taroudant and the Faculty of Applied Sciences of Ait Melloul, University Ibn Zohr.

**Institutional Review Board Statement:** Not applicable.

**Informed Consent Statement:** Not applicable.

**Data Availability Statement:** Not applicable.

**Acknowledgments:** The authors wish to thank the Laboratory of Biotechnology, Materials and Environment of the Faculty of Sciences of Agadir, the Polydisciplinary Faculty of Taroudant and the Faculty of Applied Sciences of Ait Melloul, University Ibn Zohr.

**Conflicts of Interest:** The authors declare no conflict of interest.

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
