# Peer review of "Interactive Effect of Biochar and Bio-Compost on Starting Growth and Physiologic Parameters of Argan"

_sustainability, doi:10.3390/su14127270_

Round 1

Reviewer 1 Report

The introduction is still not enough addressing the significance of this study. Are there potted studies about biochar that related to your work? You gave some general info about biochar effects on plants, but not specifically related to your work, which is a potted study.  What is the gap between your work and previous works? Why your work is necessary?

M&M: What’s the properties of biochar, soil, and bio-compost? I meant the chemical and physical properties of them such as total porosity, water holding capacity, air space, Ph, EC etc.

Author Response

Thank you for your feedback and your clarifications, you will find our answers below.

  • Introduction reformulated (novelty statement was added before the objectives)
  • Water holding capacity, Ph, EC have been added to Table 2. Biochar, soil and bio-compost chemicals analysis. The total porosity and air space are the subject of a new article that has not yet been published

Available for further information or questions

Best regards

Reviewer 2 Report

Dear authors,

thank you for resubmit your article. But, there are many previous comments that you dont consider in your new submit. The main problem for your English editing. You have to revise your MS by a native speaker or by MDPI office.

The other comments are:

1- there is no title of the first figures (trees) line 60

2- line 40: CO2 not CO2

3- line 45: you cannot start the sentence with [20,21] showed that. You have to write the name of the authors

4- line 106: the correction was not done. Give a brief detail.

5- line 124 [37] found that: again and again you cannot start the sentence with numbers, whole MS should be revised again and rephrase the sentences.

6- Line 125 [38] did not find: same previous comment

7- the reference not follow the style.

Author Response

Thank you for your feedback and your clarifications, you will find our answers below.

  • Title of the images has been inserted
  • Corrected
  • Corrected
  • Brief metson protocol description added
  • Corrected
  • Corrected
  • Corrected

Available for further information or questions

Best regards

Reviewer 3 Report

I have re-reviewed your manuscript entitled “Interactive effect of Biochar and Bio-compost on starting 2 growth and physiologic parameters of Argan”. I appreciate your efforts in revising the manuscript.  Its quality has been improved compared to the previsions version.  However, it still has some flaws which need proper consideration. Such as:

  1. The abstract is still written in generic method and have no quantitative data which can support these statements. Simply responding “Regarding our results which have been presented in a generic way, this is due to the multiple measurements in each parameter tested, which makes it difficult to cover all the major findings at the same time in the abstract” dose not justify it. You need to present few of your major finding here, which can support these statements.
  2. I suggested giving one line recommendation at the last section of abstract but I still can’t see it, why?
  3. Never use too much title words as key words.
  4. I asked for novelty in the study. If the biochar and bio-compost have already been explored and established to have so much positive effects as you are referring to. Then why you conducted this study? What was the need to repeat it? You must establish and report the answer to these questions in the last section of introduction (which I call a novelty statement).
  5. Line 110: which type ANOVA? Simple or factorial. I am still not satisfied with your answer to the question regarding the statistical design.
  6. Last time I asked, Why your results fluctuate even under control (pot) condition? You responded that this is due to the sensitivity of this tree (after several tests with other combinations between BC and CP the vast majority of combinations have died). If the tree is too much sensitive then you were supposed to conduct your experiment under very much control conditions. Your statement makes your finding ambiguous because i can say that the variation in your results might be due to experimental error not due to your treatments. Then how will you justify it?
  7. Figure 2: why your control treatment does not have SD bar? Also add the LSD value to the figure caption.
  8. I am still not satisfied by the amendments you made to the conclusion section. I suggest properly revise it.

Author Response

Thank you for your feedback and your clarifications, you will find our answers below.

1- Abstract reformulated

2- Recommendation in the last section of abstract added

3- Your remark has been taken into consideration, two words have been changed

4- Novelty statement was well mentioned on the introduction

5- Thank you for clarifying your question, We worked just with "Simple ANOVA" since the main objective of this study it is to study the effect of each treatment "Base" separated from the other regardless of the interactions that may be present between biochar and biocompost

6- Yes you are right but the only reason that prevented us from conducting our experiment under very much control conditions (in a greenhouse; temperature, humidity and luminosity controlled ...) is that we are going to be very far with the living conditions of this wild tree, which will make our results very far from reality and worthless

7- SD of control treatment is 0 because the WHC of all repetitions = 0.62 g H2O / g dw; The LSD Value added

8- Conclusion reformulated

Available for further information or questions

Best regards

Round 2

Reviewer 1 Report

Good job.

Author Response

Thank you so much

Reviewer 3 Report

Dear Author,

The changes made are fine. Now your manuscript is of sufficient quality to be published.

Author Response

Thank you so much

This manuscript is a resubmission of an earlier submission. The following is a list of the peer review reports and author responses from that submission.

Round 1

Reviewer 1 Report

The argan tree, which is found in southern Morocco, is characterized by environmental, economic, and nutritional benefits. It is necessary to find methods to accelerate the slow growth of this tree. So the starting point of this study is meaningful and the workload is also great. However, the manuscript has the following problems:

  1. The research methods are introduced too much in the abstract part, and the analysis of the research results is less.
  2. There are errors in the format of the paper, such as 105 lines, 57, lines,104 lines, etc., which need to be checked and modified in the full text.
  3. There is something wrong with the illustrations of the paper (Figures 1, 2, 3 and 4), which may be that they are not carefully checked in the process of format conversion.

Reviewer 2 Report

Dear Authors I have reviewed the manuscript entitled “The effect of Biochar and Bio-compost at different doses on starting growth and physiologic parameters of Argan” and found it novel. This article explores the interactive effect of biochar and compost on the growth of Argan. However, there are several typing and grammatical mistakes that need to be corrected. The statistics used for the experiment and the data is presentation is in confusing manner. Furthermore, there are some other technical suggestions and questions if addressed will further improve the quality of the manuscript. 1. The title shall be amended as “interactive effect of Biochar and Bio-compost s on starting growth and physiologic parameters of Argan”. 2. Line 12-13: amend the sentence as “A pot experiment was conducted to evaluate the effects of Biochar (BC) and Bio-compost (BC) each applied at the rate of 0, 3 and 6% (M/M) on starting growth of argan in fine silty soil during sixteen months”. 3. Line 14-15: Remove this sentence (After measuring the water holding capacity (WHC) and cation exchange capacity (CEC) of each mixture, chemical analyzes of soil, bio-compost and biochar were also performed) from abstract section. 4. The results presented in abstract section are generic. I suggest to present your major findings in quantitative way. 5. The introduction is too short and has no novelty statement. In this section, You need to explore that how biochar and compost is effective? Explain the mechanism. 6. Line 91: Data Analysis is also very confusing. Your results seems to be processed by simple complete randomize design. I think it would be better to run your data via two factorial CRD as you have 3 levels of each biochar and compost. Thus will have both main as well as interactive effect of both biochar and compost. 7. Table 1; The OC content of the biochar is very low, why? Usually biochar is used for C sequestration as it contains very high content of C. 8. Table 1; Never use the unit ppm, always use mg kg-1. 9. Did you measure the CEC of biochar? 10. Biochar is famous for its high CEC. Then why your soil CEC is decreased from 999 to 666 meq/100g by addition of 3% BC alone or with 3%CP. 11. Table 3: where is the main effect of biochar and compost? 12. Line 289: Do you know the nature and origin of bio-compost and you are referring to in conclusion? Revise your conclusion section. 13. What are your recommendations? Why your results fluctuate even under control (pot) condition? Did you explain the reason of fluctuation in your discussion?

Reviewer 3 Report

Dear editor,

Thank you for inviting me to evaluate this article. The effect of Biochar and Bio-compost was studied in several articles before. So, there is no novelty of this article. However, the works on Argon plant is rare which could be suitable for published after major revision.

Best wishes

Reviewer 4 Report

Dear Editor, Dear Authors.
After reading the article, I find it valuable and brings a lot of new information to science. Most likely should be published in Sustainability, but with some corrections.
After reading the article, I got the impression that this is a normal report from the research carried out. No scientific spirit, no justification for what we do this. The introduction presented by the authors is unacceptable. When writing an article about an ankai, the author should first of all justify why he is doing it ???? And it is not here and it needs to be refined.
Secondly, there is no clearly defined purpose and scope of the work. There are links to what the Authors will do, but not on a clear message.
Third, the lack of a thoroughly described experience:
- experiment scheme
- number of repetitions,
- why such a soil and not different
- why such a dose of biochar and not a different one.
These things should either be founded by the Authors or based on literature.
A very good analysis of the obtained results, thorough and detailed, and I have no objections to that. The graphic description is also very good.
Villages are another note.
There are no conclusions here, only a summary and not fully corresponding to the title of the work, because there is no specific purpose and scope.
After completing my comments, the article should be re-reviewed.

Reviewer 5 Report

Introduction:

The introduction is too short and not to the point. What is biochar? What is bio-compost? What’s the significance of this study? What’s their effects on plants, especially on potted plants since that’s your study’s focus? There are also some grammar issues. The introduction gave me the impression that the author did not read enough literature on the topic.

  • Line 29: All the scientific names of plants need to be italicized.
  • Line 34: Why suddenly changing the subject from argan tree to forest?
  • Line 41: Please define every synopsis first, CEC etc.
  • Line 41-43: no need to capitalize biochar in the middle of a sentence.

M&M:

Some major information about the substrates were mission, the experimental design was not clearly stated.  

  • Line 49: What’s the two types of biochar? The sentence describing biochar is confusing, looks like just one type. Also, although the bio-compost is purchased, what types of compost, what does it contains, which brand, which company etc. You need to give specific information so that when other scholar wants to replicate this study, they can.
  • Line 54: what is Cl? The chemical elements or the volume?
  • What is the experimental design?
  • What’s the properties of biochar, soil, and bio-compost?

Results and Discussions:

  • For some reason, figure 1-4 were not complete. Check the font size for figure 4, 7.
  • Line 99, 100: change between into among
  • Figure 5: why two legends? Please add significant differences among treatments.
  • Please check the results and discussion part, make it more concise and relevant. Compare the findings from this study with previous studies and give more explanations.

Conclusions:

  • Line 289-292: These two sentences are confusing, please rewrite.